# Occurrence and Degradation of Free and Conjugated Estrogens in a River Receiving Feedlot Animal Discharge

Hao-Shen Hung [ID], Kuei-Jyum C. Yeh, Chi-Ying Hsieh * and Ting-Chien Chen *

Department of Environmental Science and Engineering, National Pingtung University of Science and Technology, Pingtung 91201, Taiwan
* Correspondence: chiying@mail.npust.edu.tw (C.-Y.H.); chen5637@mail.npust.edu.tw (T.-C.C.);
  Tel.: +886-8-774-0425 (T.-C.C.); Fax: +886-8-774-0122 (T.-C.C.)

**Abstract:** This study analyzed concentrations of 17β-estradiol (E2), estrone (E1), estriol (E3), 17α-ethynylestradiol (EE2), diethylstilbestrol (DES), 17β-estradiol-3-sulfate (E2-3S), estrone-3-sulfate (E1-3S), 17β-estradiol-3-glucuronide (E2-3G), and estrone-3-glucuronide (E1-3G) in river water, received from intensive feedlot operations wastewater in WuLo Creek, Taiwan. Moreover, the estrogen degradation in situ was analyzed. The average concentrations were $54.15 \pm 31.42$, $9.71 \pm 6.42$ and $3.55 \pm 2.41$ ng/L for E1, E2 and E3, respectively. The concentrations and order were similar to the polluted river and higher than most rivers' concentrations. The conjugated estrogen concentrations ranged from ND to 13.2 ng/L (E1-3S), ND to 10.4 ng/L (E2-3S), ND to 10.0 ng/L (E1-3G), and ND to 3.6 ng/L (E2-3G), and the detection rates were 76%, 71%, 56%, and 15%, respectively. In the present study, the high detection rates of conjugate estrogen were more elevated than the water receiving STP effluent, suggesting that the source was the river water close to the animal wastewater discharge. In the degradation test, the DES concentrations slightly declined after 24 h, but E1-3G did not significantly change over time ($p > 0.05$). The degradation of free estrogen occurred during the first 12 h period, and residual concentration was not further decreased after 24 h. In the environment, E1 had higher concentrations than E2 and E3, suggesting that E1 was more resistant to degradation than E2 and E3 at low concentrations. However, the degradation test in the present study suggested that E1 rapidly degraded at high ambient concentrations due to the high degradation constant.

**Keywords:** free estrogen; conjugate estrogen; estrogen degradation; first-order kinetic constants; half-life; environmental pollution

## 1. Introduction

Many endocrine-disrupting compounds (EDCs) have been found in water environments. However, they interfere with the endocrine system and cause possible adverse effects in aquatic organisms at low concentrations (ng/L) worldwide [1–5]. For example, 1 ng/L 17β-estradiol (E2) in a water environment induces vitellogenin production in male trout [3] and it caused intersex (i.e., testis-ova) and altered sex in Japanese medaka from hatching to approximately 100 days after hatching [6].

Estrogen compounds are one of the main EDCs, which humans and animals discharge. The most commonly detected species include free estrogen, such as E2, estrone (E1), and estriol (E3), and synthetic estrogens, such as 17α-ethynylestradiol (EE2), and diethylstilbestrol (DES) [7,8]. Conjugated estrogens have less biological effectiveness but can convert to free estrogen during dissociation or microbial conversion to increase their efficacy [9–11]. However, that is often ignored due to its low detection rate and concentration, low estrogen potency, and short half-life in the environment [9,10,12–16].

There are several emission sources for estrogen into the environment, such as sewage treatment plants [17–21], industrial wastewater [22,23], and water bodies receiving urban wastewater. However, there is little data on the estrogen concentration from intensive

animal feedlot operation discharge compared with the water bodies affected by human discharge [1,24,25]. An intensive animal feedlot dramatically increases the discharge estrogen concentration. Previous studies suggested that animal estrogens are ubiquitous in the environment, resulting from manure input or uncontrolled natural sources [12,26]. Animal excrete contains many estrogen compounds that promote estrogenic potency in the receiving water environment, which has a high variation proportion of conjugated estrogen from 3% to 95% [12,27–30] and about 73% to 83% of estrogen estimated from animal excrement into an environment [25,31]. Remarkably, the presence of estrogen in rivers endangers ecological safety, so more information on the estrogen fate in rivers is needed. Hence, estrogen in animal discharge requires further investigation of the fate in the aquatic environment.

Degradation is one of the main factors affecting estrogen's fate in an aquatic environment. Degradation half-life is used to measure the stability and persistence of estrogen substances in the environment. Some previous studies have focused on laboratory addition experiments to know which microorganisms contributed to the degradation of estrogens in the aquatic environment [14,32]. Although estrogen can be degraded in the laboratory, the studies used a high concentration addition of a single estrogen in river water or a few species of estrogens [33,34]. Previous studies showed the degradation half-life of E1 and E2 was between 0.1–11 and 0.2–9 days, respectively [8,33]. Kumar et al. [21] pointed out the half-life of Glu-type estrogen was 0.4 h in sludge and increases to 2 days in the river water. With the spiking experiment, the initial 17β-estradiol-3-sulfate (E2-3S) concentration was from 40 to 4000 ng/L, and the half-life was from 2.7 to 86.6 h [35]. Those methods still cannot reflect river water's actual degradation because of the matrix's influence and the existence of reversible reactions of various compounds [4,33–35]. The release of steroid hormones into the river from animal feedlot discharge is usually associated with a high carbon content of manure, manure-contaminated wastewater, or other biosolids. To our knowledge, there is no study on the degradation of estrogen in river water. In addition, an intensive feedlot may dramatically increase high estrogen concentrations from a few ng/L to μg/L in the receiving water body [24,34]. Many factors, such as temperature, pH, aerobic, anaerobic, liquid properties, organic matter, microbial species, and initial estrogen concentration, in addition to a degradation experiment, can affect the degradation potency [4,10,17,32,35–37]. However, degradation fate has more complex conditions in the natural environment than in the laboratory and may induce different results.

Changes in river characteristics due to intensive feedlot operations may significantly impact estrogen degradation. The addition to the batch experiment found estrogen degradation at the start time and followed persistently at low concentrations in 28 d [10]. This study investigated estrogen degradation in natural river water, sampling from Wulo Creek, where intensive feedlot operation discharges were received. Chen et al. [24] reported high estrogen concentrations in the river water. Therefore, the objective of this study was to investigate estrogen concentrations and the kinetics of actual degradation in the river water receiving intensive feedlot operation discharges. The degradation constant and half-life of detected estrogens will be compared to laboratory batch experiments. The study will provide further data and fill the knowledge gaps about the degradation of estrogen from an intensive feedlot operation in the receiving river.

## 2. Materials and Methods

### 2.1. Sampling Method and Location

Wulo Creek is a tributary of the Gaoping River, and its watershed covers Jiuru Township and Yanpu Township, where animal husbandry is relatively intensive in Pingtung County, Taiwan. The livestock breeding capacity in this area is about 4500 cattle, 500,000 pigs, 3 million laying hens, and 5 million broilers. This sampling location refers to the high-estrogen-concentration regions in the six stations from Chen's study [24] (Figure 1).

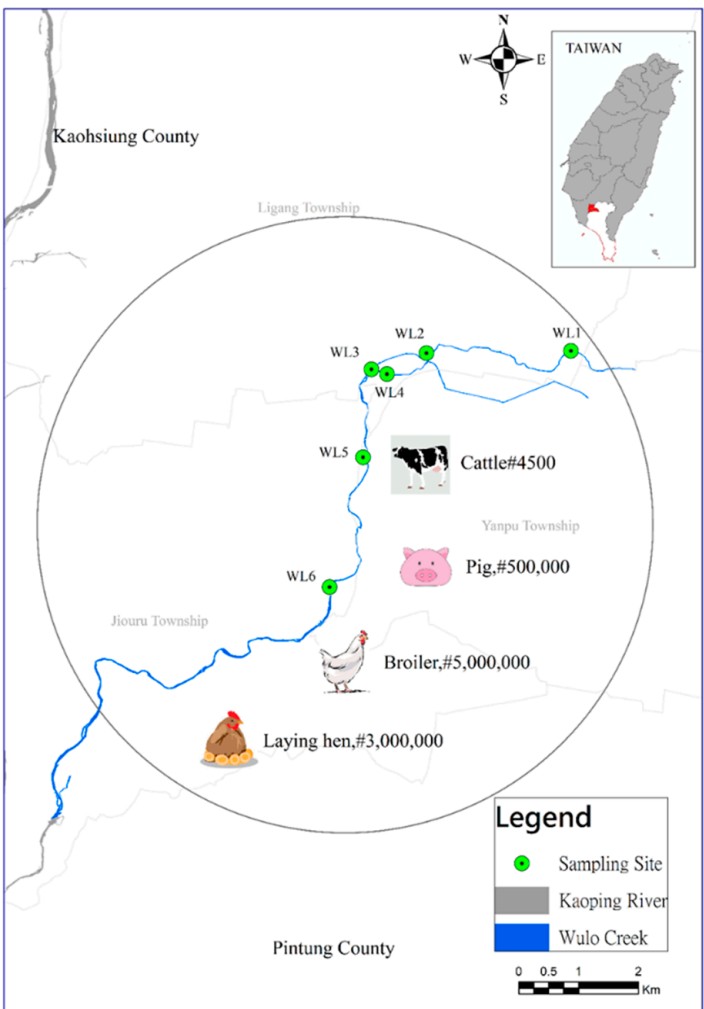

**Figure 1.** Sampling locations of Wulo Creek.

Animal farms have different cleaning frequencies in different seasons, e.g., in summer, pig farms are cleaned every day; in winter, pig farms are cleaned every other day. The wastewater from cleaning the pig farm, usually through simple wastewater treatment procedures, removes the pollution, including solid–liquid separation, equalization, anaerobic, aerobic, and settling procedures. Chicken manure is cleaned once every two months, and the chicken manure is dried to keep the moisture content below 25%. After drying, the dried manure is generally applied to the farmland as fertilizer. Previous studies [24] show that runoff contains high estrogens concentrations because of using manure on the farmland. In addition, when it is discharged into the river, it can impact the estrogen concentrations [38,39].

### 2.2. Water Quality Parameter

Water quality parameters were directly analyzed on-site, including temperature, pH (WTW, pH315i/SET), conductivity (EC, WTW, LF330/SET), and dissolved oxygen (DO, WTW-Oxi, 3205SET). Filtered water samples (1.6 μm, Pall) measured ammonia (NH$_3$-N, HANNA, HI 93715). Standard Methods analyzed alkalinity, total solids (TS), and suspended solids (SS). DO is regarded as the quality of water with high concentration, the better the water quality. TS is the total amount of solids dissolved in the water, which affect the taste of drinking water. SS is suspended solid matter in water, which is one of the indicators to measure the degree of water pollution. Chemical oxygen demand (COD) measures water quality by chemical oxygen consumption while the decomposing organic matter is in the

water. This study used COD analysis by the Hach method. A reagent (Hach 21259-25) was added to the water sample, then heated at 150 °C for two h. After cooling, the water sample was measured with a spectrophotometer (Hach DR/2010). Dissolved organic carbon (DOC) is defined as dissolved organic matter that can pass through a 0.45 μm filter and be lost by evaporation during the analysis process.

*2.3. Pretreatment of the Estrogen-Containing Water Samples*

The water samples were subjected to fine filtration (0.22 μm) using cellulose acetate membrane materials to remove suspended solids. Then, filtered water samples used sulfuric acid to acid-attain pH 3.0; the water samples were stored at 4 °C until analysis. The target compound was concentrated by passing solid-phase extraction (SPE) cartridges using Oasis HLB (200 mg/6 mL, 30-μm partial size, Waters), combined with Sep-Pak and $NH_2$ (360 mg, aminopropyl, 55–105 μm partial size, Waters), which were pre-conditioned with 6 mL of methanol and 6 mL of deionized water. A total of 1000 mL of the acidified (pH 2–3) water sample was passed through the SPE column at a 3–6 mL/min flow rate. Subsequently, 5 mL of 5% methanol was used to elute the cartridges and then dry them under a vacuum for 30 min. After that, the column was eluted with 3 mL of 5% methanol/double-distilled water (DDW). Next, 6 mL of methanol was used to elute the column. Finally, the cartridge was eluted after 0.5% $NH_4OH$ with 6 mL 2% acetonitrile/methanol (ACN/MeOH). The eluted sample was evaporated with slowly blown nitrogen to achieve a near-dryness state and redissolved with 2 mL of the ACN/DDW ($v/v$ = 10/90) solution. Finally, it was passed through a 0.22-μm polytetrafluoroethylene syringe filter for LC/MS/MS analysis. This method was modified from previous studies to get the best parameters [30,40–42].

*2.4. LC/MS/MS Analysis Conditions*

2.4.1. Instrumental Analysis

The Agilent 1200/6400 quadrupole LC/MS/MS system is equipped with an electrospray ionization (ESI) interface to analyze estrogen concentration (Agilent Technologies, Palo Alto, CA, USA). The separation was performed using Phenomenex Gemini C-18 Column (100 × 2.0 mm, dp = 3 μm) at 40 °C. The mobile phase consisted of water (A) and acetonitrile (B) both containing 0.05% ammonium hydroxide. The programmed gradient initial flow rate of 0.2 mL/min was maintained at 20% of B for one minute, then increased to 35% of B in 1.5 min, rose to 58% of B in 1.5 min, and then maintained for 0.3 min. At 6.7 min, B increased to 85%; the total analysis time was 11 min. The negative polarity ionization mode was operated to obtain the best mass spectra (standard hormone solutions) for the identification of transformation products of compounds. The operating conditions for ESI were capillary voltage of 4000 V for negative electricity to spray the ion source, drying gas flow rate of 8 L/min at 325 °C, and nebulizer gas pressure of 30 psi.

2.4.2. Quality Assurance and Quality Control

The calibration curve for each estrogen-containing sample exhibited strong linearity ($R^2 > 0.995$) over a range of concentrations (ng/L in samples). The method detection limit for the selected estrogen species ranged from 0.3 to 0.6 ng/L. Table S1 lists physicochemical properties of steroid estrogens. The recoveries were determined using a standard addition method at a spiking concentration of 10 ng/L for the river water. The corresponding recovery rates were in the range of 75–107%, respectively. Table S2 lists quality assurance data for analyzing each target compound in DI water and river water.

2.4.3. Statistics

SPSS software was used for statistical analysis. The Gaussian distribution of continuous variables was tested by the Shapiro–Wilk (S-W) and Kolmogorov–Smirnov (K-S) test methods. Furthermore, the linear regression was performed by simple regression analysis.

## 3. Results

### 3.1. Water Quality

The water quality parameter statistic summaries are listed in Table 1. The basin is located in a subtropical area, and the water temperature was between 20.0–33.5 °C, with an average of 27.9 °C. The basin contains large amounts of animal wastewater discharge. Therefore, it is classified as heavily polluted water in terms of water quality, such as high ammonia nitrogen (mean 7.3 ± 5.3 mg/L) and dissolved organic carbon (DOC average 9.3 ± 5.9 mg/L), low dissolved oxygen content (DO average 2.3 ± 1.3 mg/L). In addition, the average total solids (TS, 815 mg/L) and suspended solids (SS, 106 mg/L) differed up to 709 mg/L, suggesting a filtered water sample containing a high concentration of dissolved organic matter. Therefore, the high dissolved organic matter and ammonia concentrations in the samples revealed the features of agricultural wastewater.

**Table 1.** Water quality parameters of samples.

|  | T | pH | EC | DO | NH$_3$-N | SS | TS | DOC |
|---|---|---|---|---|---|---|---|---|
|  | °C |  | µS/cm | mg/L | mg/L | mg/L | mg/L | mg/L |
| Mean | 27.9 | 7.3 | 698 | 2.3 | 7.3 | 106 | 815 | 9.3 |
| SD | 3.3 | 0.3 | 248 | 1.3 | 5.3 | 68 | 473 | 5.9 |
| Max | 33.5 | 8.3 | 1719 | 7.6 | 24.8 | 343 | 2867 | 22.6 |
| Min | 20.0 | 6.8 | 14 | 0.0 | 1.5 | 25 | 441 | 2.3 |
| Median | 29.0 | 7.3 | 725 | 2.1 | 5.8 | 86 | 660 | 7.8 |

### 3.2. Estrogen Concentrations in Water Samples

This study conducted the sampling campaign from 2011/01 to 2012/02 at six sites (WL1 to WL6). Samples were taken 14 times at sampling sites (WL1 and WL4) and other sites 13 times. In total, 80 samples were collected, in which five samples failed to be extracted; estrogen concentrations could not be detected. Therefore, 75 valid samples were analyzed for estrogen concentrations. In this study, the detection rate indicated that each estrogen concentration was greater than the statistical quantity of MDLs, and the non-detectable concentrations were ignored. Table 2 lists the estrogens' statistical summaries, including concentration range, detection rate, and *p*-value for the normality test. Figure 2 shows the average concentration and median for each species. The normality test had the same results for both test methods. The estrogen concentrations are not in the normal distribution ($p < 0.05$) for E2, DES, EE2, E1-3G and E2-3G. The E1, E3, E1-3S and E2-3S concentration had a normal distribution.

**Table 2.** Estrogen concentrations ranges, detection rates and *p* values of the normality test.

| Compounds | Concentration Range (ng/L) | Detection Rate (%) | *p*-Value * | *p*-Value ** |
|---|---|---|---|---|
| E1 | 2.0–145.6 | 100 | 0.169 | 0.200 |
| E2 | ND–13.2 | 84 | 0.003 | <0.001 |
| E3 | ND–34.0 | 95 | 0.083 | 0.200 |
| DES | ND–10.0 | 63 | <0.001 | <0.001 |
| EE2 | ND–4.4 | 15 | 0.001 | <0.001 |
| E1-3S | ND–13.2 | 76 | 0.455 | 0.131 |
| E2-3S | ND–10.4 | 71 | 0.053 | 0.098 |
| E1-3G | ND–10.0 | 56 | 0.001 | 0.005 |
| E2-3G | ND–3.6 | 15 | 0.005 | <0.001 |

Normality test for the concentration < MDL was ignored; * first *p*-value Shapiro–Wilk (S-W) test method; ** second *p*-value using Kolmogorov–Smirnov (K-S) test method.

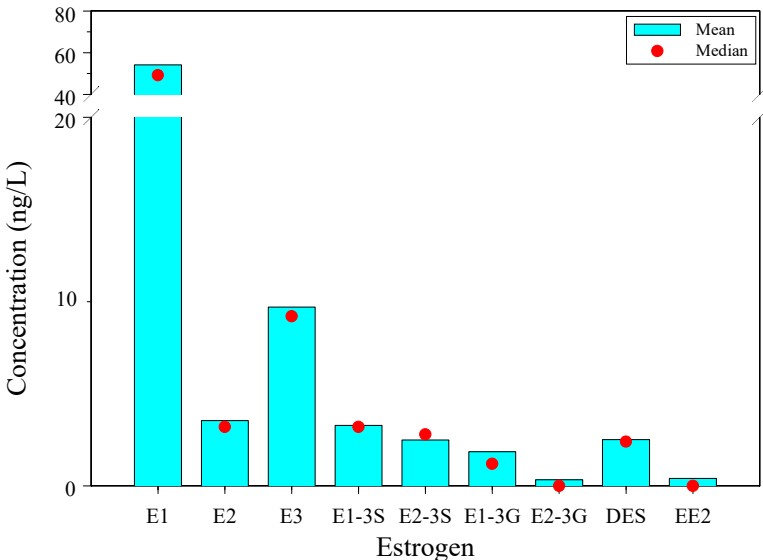

**Figure 2.** Estrogen mean concentration and median values.

### 3.2.1. Concentration of Free Estrogen

Free estrogen concentrations varied greatly. E1 had the highest concentrations, ranging from 2.0 to 145.6 ng/L, the detection rate was 100%, and the median concentration was 49.2 ng/L. Following this was E3, which ranged from ND to 34.0 ng/L, the detection rate was 95%, and the median concentration was 9.2 ng/L. The lowest concentrations were seen for E2, ranging from ND to 13.2 ng/L, the detection rate was 84%, and the median concentration was 3.2 ng/L. The average concentration of free estrogen was $54.2 \pm 31.4$, $9.7 \pm 6.4$ and $3.6 \pm 2.4$ ng/L for E1, E3 and E2, respectively.

In the synthetic estrogen DES and EE2 samples, DES concentrations were higher than EE2 concentrations. DES concentrations ranged from ND to 10.0 ng/L, and the detection rate was 63%. The median concentration was 2.4 ng/L. EE2 concentration ranged from ND to 4.4 ng/L. EE2 is the main component of contraceptives, the average concentration was 0.4 ng/L, and the detection rate was 15%. DES average concentration was $2.5 \pm 2.5$ ng/L, and the detection rate was 63%.

### 3.2.2. Concentration of Conjugated Estrogens

This study analyzed four species of conjugated estrogen, including E1-3S, E2-3S, E1-3G, and E2-3G, which were lower than the free estrogens concentrations E1 and E3 concentrations but similar to concentrations of E2. The following concentrations of conjugated estrogens were E1-3S, E2-3S, E1-3G, and E2-3G, with concentrations ranging from ND to 13.2 ng/L, ND to 10.4 ng/L, ND to 10.0 ng/L, and ND to 3.6 ng/L, respectively. The detection rates were 76%, 71%, 56%, and 15%, and the median concentration was 3.2 ng/L, 2.8 ng/L, 1.2 ng/L, and ND, respectively.

### 3.3. Degradation of Estrogen in the Water Sample

To investigate the degradation characteristics of estrogens in river water, 20 L of water was collected from six sites on 28 October 2011. Estrogen concentrations were analyzed at 0, 3, 4, 6, 8, 12, and 24 h for each water sample. The concentrations of E2-3G, E1-3S, and E2-3S are below the detected limitation and not discussed in the text. Table 3 lists the initial estrogen concentrations ($C_0$) and concentrations in each water sample at 24 h ($C_{24}$). The initial concentrations of samples were measured for five kinds of estrogen species, DES, E1, E2, E3, and E1-3G. Observed from the beginning and 24 h estrogen concentration (Table 3), only free estrogens E1, E2, and E3 concentrations significantly decreased, DES concentration slightly decreased, and E1-3G concentration did not decrease.

**Table 3.** Initial estrogen concentrations ($C_0$) and concentrations at 24 h ($C_{24}$) of target estrogens for water sample (ng/L).

|  | DES | E1 | E2 | E3 | E1-3G |
|---|---|---|---|---|---|
| **Site** | $C_0$–$C_{24}$ | $C_0$–$C_{24}$ | $C_0$–$C_{24}$ | $C_0$–$C_{24}$ | $C_0$–$C_{24}$ |
| WL1 | 2.8–2.4 | 20.4–4.0 | 3.6–2.4 | 4.0–2.0 | 2.0–2.8 |
| WL2 | 3.6–2.0 | 67.6–5.6 | 4.4–2.4 | 7.2–2.8 | 3.2–3.6 |
| WL3 | 6.2–3.2 | 145.6–3.2 | 6.4–3.6 | 13.6–2.0 | 2.0–4.0 |
| WL4 | 2.4–2.4 | 41.2–9.6 | 2.8–2.0 | 7.2–2.8 | 2.8–3.6 |
| WL5 | 2.0–2.4 | 19.6–4.0 | 2.8–2.0 | 3.6–2.0 | 2.8–4.0 |
| WL6 | 2.8–2.4 | 14.0–5.6 | 2.4–2.0 | 2.0–2.0 | 2.0–2.4 |

Free estrogen concentrations, changing over time, were simulated with a first-order kinetic equation (Equation (1)), which was applied to simulate free and conjugated estrogens degradation reactions [16,33,43–47]. Figure 3 shows average estrogen concentration ratios ($C_n$/$C_0$) over time and line regression curves. Table 4 lists the linear regression parameters including slope, r value, and *p* value from 0–12 h. The *p* values ($p < 0.05$) show E1, E2, and E3 had significantly decreased, while DES and E1-3G had insignificantly decreased. Moreover, DES had a median r value (r = 0.59) and a negative slope suggesting DES slightly decreased over time. However, E1-3G had a small positive slope and r value > 0.5, suggesting that E1-3G did not change over time.

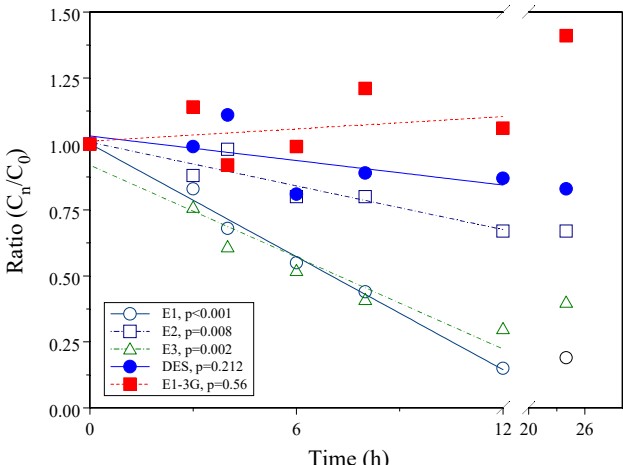

**Figure 3.** Estrogen concentration ratio ($C_n$/$C_0$) changed with time and linear regression curves for the five detected compounds.

**Table 4.** The parameters of linear regression of average $C_n$/$C_0$ ratios changed with time for the five detected compounds.

| Parameters | DES | E1-3G | E1 | E2 | E3 |
|---|---|---|---|---|---|
| Slope | −0.016 | 0.0078 | −0.0714 | −0.0276 | −0.058 |
| Intercept | 1.030 | 1.011 | 1.009 | 1.007 | 0.9093 |
| r | 0.60 | 0.31 | 0.996 | 0.927 | 0.963 |
| *p*-value | 0.212 | 0.56 | <0.001 | 0.008 | 0.002 |
| F1 | 2.20 | 0.41 | 473.06 | 24.47 | 8.34 |
| F2 | 1.10 | 0.67 | 0.05 | 0.26 | 1.64 |

$p < 0.05$ indicated the data set had a significantly linear relationship. F1: F-test value is $MS_{REG}/MS_R$ (Mean square of the regression/Mean square of the residuals). F2: F-test value is $MS_{LOF}/MS_{PE}$ (Mean Square of lack of fit/Mean Square of pure error).

Figure 4 shows three free estrogen average concentrations (ln $C_n$/$C_0$) changed with time and shows linear regression of three kinds of free estrogen concentration (ln $C_n$/$C_0$)

versus time (0–12 h). The kinetic rates of degradation (slopes of regression) for free estrogens at each site are listed in Table 5. Equation (2) calculated the half-life of degradation based on the regression slope. The half-life box plot of three free estrogens in the six sites is shown in Figure 5.

$$\ln\left({C_n}/{C_0}\right) = -k \cdot t \tag{1}$$

$$T_{1/2} = {0.693}/{k} \tag{2}$$

where $C_n$ was the estrogen concentrations at analyzed times 3, 4, 6, 8, 12, and 24 h, and $C_0$ was the initial estrogen concentrations at 0 h. k is the first-order kinetic constant ($h^{-1}$). t is the time (h), and $T_{1/2}$ is the half-life (h).

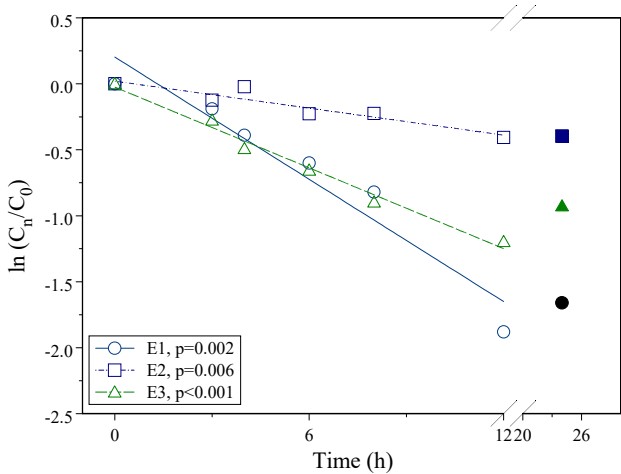

**Figure 4.** Estrogen concentration ratio ln (Cn/C₀) changed with time and linear regression curves for E1, E2, and E3.

**Table 5.** The parameters of linear regression of average $\ln C_n/C_0$ changed with time for E1, E2 and E3.

| Parameters | E1 | E2 | E3 |
|---|---|---|---|
| Slope | −0.154 | −0.034 | −0.102 |
| Intercept | 0.203 | 0.0197 | −0.025 |
| r | 0.964 | 0.937 | 0.99 |
| *p*-value (R) | 0.002 | 0.006 | <0.001 |
| F1 | 52.15 | 29.10 | 9.52 |
| F2 | 0.20 | 0.16 | 0.67 |

$p < 0.05$ indicated the data set had a significantly linear relationship. F1: F-test value is $MS_{REG}/MS_R$ (Mean square of the regression/Mean square of the residuals). F2: F-test value is $MS_{LOF}/MS_{PE}$ (Mean Square of lack of fit/Mean Square of pure error).

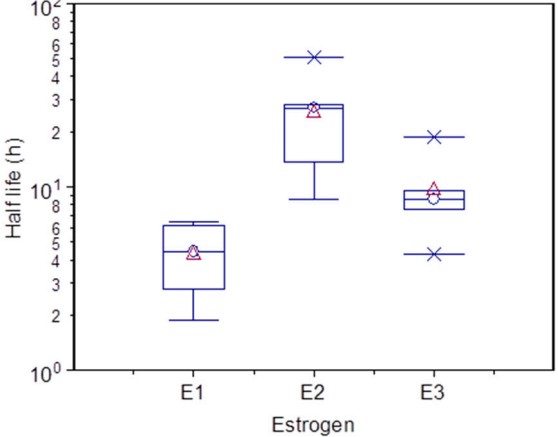

**Figure 5.** The boxplot of half-life of three free estrogens.

## 4. Discussion

### 4.1. Estrogen Concentrations

The average concentrations of free estrogen were $54.2 \pm 31.4$, $9.7 \pm 6.4$ and $3.6 \pm 2.4$ ng/L for E1, E3 and E2, respectively, which is similar to a polluted river and higher than most rivers' concentrations [48–52]. The free estrogens concentration order is similar to water receiving the discharge of animal waste and wastewater [48–52]. The EE2 concentrations and detection rates were lower than in most rivers receiving sewage plant discharge at EE2 concentrations ranging from ND to 58 ng/L [53–55]. In addition, the EE2 average concentrations of WWTP effluents arrived with a hundred ng/L in high population areas [55]. DES is mainly used as an additive to promote the growth of animal feeding. Although DES has been banned, the abuse of DES to promote animal growth continues [7]. The water body that received animal wastewater discharge still detected DES in water samples [55,56]. Additionally, a previous study detected EE2 and DES in fish tissues around the polluted river [57].

Similar to this study, Yu et al. [4] reviewed the study summary of 29 rivers and 217 sites, and concentrations of the E1-3S, E2-3S, and E1-3G ranged from ND (0.013)-14.0 ng/L, ND (0.1)-26.3 ng/L, ND (0.1)-0.83 ng/L. The study explains that conjugated estrogen E2-3G is degraded more quickly in the river; previous research shows only one concentration is detected above ND [4]. Kumar et al. [58] study also showed that only one glucuronide conjugate could be found in the eleven sewage treatment plants effluent, so this study suggested that the intensive feedlot operation increased the detected frequency and concentrations of conjugated estrogens [4,58,59].

### 4.2. Degradation of DES and E1-3G

This study investigated the degradation of estrogens in in situ river water. Theoretically, the concentration of estrogen will decrease with time if the degradation occurs; hence, the estrogen concentration ratio ($C_n/C_0$) changed with time, having a negative slope and a *p*-value of <0.05 in the linear regression. Figure 3 shows the degradation curves of DES, E1-3G, E1, E2 and E3. The linear regression was statistically tested and the results are listed in Table 4. The linear results show E1-3G had a positive slope value (0.0078) and insignificantly decreased (r = 0.31, *p* = 0.56). DES had a negative slope value ($-0.016$) and insignificantly decreased (r = 0.60, *p* = 0.21). The E1, E2, and E3 had negative slope values and significantly decreased (*p* < 0.05) as shown in Table 4. The negative slope value and *p* < 0.05 suggests the concentration of E1, E2 and E3 significantly decreased over time. Conjugated estrogen E1-3G had almost no degradation in this study. The initial concentrations of E1-3G ranged from 2.0 to 3.2 ng/L, representing a relatively low concentration. E1-3G can be regarded as residual concentrations and had been through a fast dissociation step after it was released into the river, thus demonstrating slow degradation. This is similar to the Ma and Yates [35] study where additional experiments monitored the E2-3S and E2-3G at every two concentrations of 40 and 4000 ng/g for estrogen degradation and metabolite formation in agricultural soil. At an initial 24 h, more than 95% was removed and reached a steady-state after 36–72 h, with persistence levels remaining after 20 days. Similar results found that the aqueous phase persisted for up to 28 d; E2-3G had additional concentrations of 3.7, 9.1, and 22.5 µg/L [10]. Other possible reasons included the river water lacking a β-glucuronidase enzyme to dissociate Glu-ligand estrogen and the occurrence of reversed reaction between E1-3G and E1 [13].

Unlike in laboratory experiments, natural intensive feedlot operation receiving environments cannot control experimental factors; they are more complex and continue to be polluted. Compared to the DOC of this study, the estrogen concentration was lower than other carbon sources; at low concentrations, the microorganism may prefer to use the other carbon source [60]. For E2 degradation experiments, previous studies indicated that bacterial strains (*Ochrobactrum* sp. *Strain FJ1*, E2 degrading bacteria) preferred to use the additional carbon source instead of estradiol [11]. On the other hand, intensive feedlot operations increased the number of veterinary antibiotics used, affecting estrogen persistence

when released to the aquatic environment. The previous study indicated that veterinary antibiotics used for an animal could affect E2 persistence in swine manure [61]. Therefore, the E1-3G concentrations in the degradation test did not change in 24 h. Additional experiments monitored the E1-3G concentration of 40 and 4000 ng/g for half-lives, 0.81 and 8.56 $h^{-1}$, respectively [35]. This study may mean E1-3G was present in the intensive feedlot operation receiving river environment and may have reached a steady-state at low concentration [34,35].

### 4.3. Degradation of Free Estrogen

From the results in Figure 3 and Table 4, E1, E2 and E3 had significantly decreased over time. According to Equation (1), Figure 4 shows that the average estrogen concentration ratio ln $(C_n/C_0)$ changed with time and displayed a linear regression for E1, E2, and E3. The linear regression was statistically tested and the results are listed in Table 5. For each site, the statistical results of ln (Cn/C0) changing with time are listed in Table 6. The linear regression of the average ln $(C_n/C_0)$ over time had a statistically significant linear relationship ($p = 0.006$–0.001) for E1, E2 and E3 (Figure 4). Figure 4 shows that concentrations of free estrogen rapidly decreased during the initial 12 h period, but average concentrations were not further reduced between the 12 and 24 h period. Since the rapid concentration decreased in the initial 12 h, it suggested that the microbial population had adapted to the river water environment. Thus, microbes can effectively degrade the relatively high concentrations of free estrogen in the river. Because estrogen concentrations were not further degraded during the 12 to 24 h period, the degradation rate constants were taken from regression data of concentration degradation during 0–12 h. The E1 first-order rate constants (k) were between 0.11 and 0.37 $h^{-1}$ (Table 6), and the k value was 0.154 $h^{-1}$ for the average concentration of the six stations (Table 5). The E2 rate constants were between 0.01 and 0.08 $h^{-1}$, and the k value was 0.034 $h^{-1}$ for the average concentration of the six stations. The E3 rate constants were between 0.04 and 0.16 $h^{-1}$, and the k value was 0.102 $h^{-1}$ for the average concentration of 5 stations.

**Table 6.** The kinetic rates of degradation for free estrogens at sampling sites.

| | **E1** | | | **E2** | | | **E3** | | |
|---|---|---|---|---|---|---|---|---|---|
| **Site** | **k ($h^{-1}$)** | **$R^2$** | ***p*** | **k ($h^{-1}$)** | **$R^2$** | ***p*** | **k ($h^{-1}$)** | **$R^2$** | ***p*** |
| WL1 | 0.112 | 0.55 | 0.093 | 0.026 | 0.42 | 0.168 | 0.081 | 0.63 | 0.108 |
| WL2 | 0.251 | 0.96 | <0.001 | 0.051 | 0.56 | 0.087 | 0.092 | 0.77 | 0.051 |
| WL3 | 0.371 | 0.97 | <0.001 | 0.081 | 0.86 | 0.007 | 0.162 | 0.88 | 0.006 |
| WL4 | 0.107 | 0.45 | 0.142 | 0.014 | 0.13 | 0.486 | 0.072 | 0.76 | 0.023 |
| WL5 | 0.183 | 0.98 | <0.001 | 0.026 | 0.75 | 0.026 | 0.037 | 0.46 | 0.140 |
| WL6 | 0.138 | 0.71 | 0.035 | 0.025 | 0.27 | 0.286 | NA | NA | NA |

NA: Not available.

The slopes of the average concentration of the three free estrogens are shown in Figure 4. The E1 half-lives ($T_{1/2}$) ranged from 1.9 to 6.5 h, and the average half-life was 4.5 h. The E2 half-lives ranged from 8.6 to 49.5 h, and the average half-life was 20.4 h. The E3 half-lives ranged from 4.3 to 18.8 h, and the average half-life was 6.8 h. Figure 5 shows a box-plot of half-life for the free estrogens E1, E2, and E3, and the figure also shows the median half-life values were 4.4, 27.0 and 8.6 h for E1, E2, E3, respectively. The $T_{1/2}$ values of the average and the median half-lives were very close for the free estrogens.

Compared to laboratory experiments, the field was more complex, and many factors can affect estrogen degradation in the environment [32]. A similar, previous review study on nonylphenol microbial degradation from contaminated environments showed that the k value had significant variation from 0.02 to 0.8 day$^{-1}$ at the different mediums of seawater, sediment, activated sludge, wastewater, and soil [32]. The estrogen half-life differs in the environment setting and depends on oxygen availability and degradation rate [1,4,48]. For estrogen soil research under aerobic microorganism degradation, the half-lives ranged

from 2.8–4.9, 0.8–1.1, and 0.7–1.7 days for E1, E2, and E3, respectively [62]. Half-lives were recorded in river water under aerobic environments, 2–3 days for E1 and 1–4 days for E2 [33,48]. Estrogen half-lives in biological wastewater treatment were quantified as being within 5.5 h for E1, 4.6 h for E2, and 4.9 h for E3 under aerobic conditions with activated sludge inocula [63]. In sediment addition experiments, with E2 under aerobic and anaerobic conditions, the anaerobic E2 was more persistent, with half-lives from 5.0 to 20.6 d [5]. In our study, the E2 half-lives were close to the previous river research but unlike E1 and E3 [33,42].

Estrogen concentration levels in the initial stages could affect degradation and transformation ability [35,54]. For example, microcosm laboratory experiments show that E1-3G had high half-lives and low k values at high concentrations, and the value discrepancy was 0.81, 8.56 h, and 0.08, 0.86 h$^{-1}$, which had two different E2-3G concentrations of 40 and 4000 ng/g [35]. Until the 20-d incubation, with the detectable concentration of E1-3G, albeit at a trace level (1.9 ng/L), high initial applied concentrations of the parent compounds are high [35]. The highest initial concentration of waste in dairy farms was E2, and E2 tends to degrade and transform into E1, followed by a higher concentration of E1 (i.e., E1 > E2) [64]. In the present study, the k value is slightly different and may indicate various effluent sources.

## 5. Conclusions

The free estrogen concentrations were similar to the polluted river but higher than most rivers' concentrations. Conjugated estrogens E1-3S, E2-3S, and E1-3G had high detection rates that suggested that the water samples were collected close to the discharge source. Hence, a high rate of conjugated estrogen was detected. In addition, conjugated estrogen concentrations were relative to the free estrogen E2 and E3, consisting of residual conjugated estrogens resistant to degradation. The concentration order of the free estrogens was E1 > E3 and E2, which was inconsistent with the concentration order found in most rivers but consistent with the concentration order of estrogen receiving an animal source. Because upstream samples received a discharge from animal sources, those samples had a high E3 detection rate and a concentration higher than E2. The degradation test showed that the residual concentrations of E2 and E3 had similar concentrations after 12 h of degradation, demonstrating a faster degradation rate of E3 than E2. Degradation of DES and E1-3G was slower than the free estrogens, and the residual concentration of E1-3G showed almost no degradation, which was similar to the recently published conclusions [34,35]. The degradation study of free estrogen showed that degradation occurred during the first 12 h period, and the residual concentration was not further decreased from 12 to 24 h. The average half-life ranged from 4 to 20 h. In this study, E1 concentrations were much higher than concentrations of E2 and E3, and degradation rate constants were E1 > E3 > E2. High concentrations of E1 were found in the environment. The present study suggested that E1 is more resistant to degradation. Nevertheless, E1 had a small degradation constant and a short half-life, suggesting that E1 rapidly degraded at high ambient concentrations.

**Supplementary Materials:** The following supporting information can be downloaded at: https://www.mdpi.com/article/10.3390/app122311961/s1, Table S1: Physicochemical properties of steroid estrogens.; Table S2: Quality assurance data for analyzing each target compound in DI water and river water. Reference [65] is cited in the Supplementary Materials.

**Author Contributions:** H.-S.H.: Writing, Methodology, Data curation, Visualization, Investigation. K.-J.C.Y.: Methodology, Supervision, Funding acquisition. C.-Y.H.: Methodology, Validation. T.-C.C.: Writing, review and editing, Conceptualization, Methodology. All authors have read and agreed to the published version of the manuscript.

**Funding:** This research was funded by the National Science Council of Taiwan, ROC, under grants NSC101-2621-M020-007, NSC101-2621-M020-008 and MOST107-2221-E-020-004-MY3.

**Institutional Review Board Statement:** Not applicable.

**Informed Consent Statement:** Not applicable.

**Data Availability Statement:** Not applicable.

**Acknowledgments:** The authors would like to thank the "Center for Agricultural and Aquacultural Product Inspection and Certification of National Pingtung University of Science and Technology, Taiwan" for assisting in developing the analytical methods. The LC/MS/MS used for estrogen analyses was financially supported by the National Pingtung University of Science and Technology. This study also thanks the colleagues for assisting the experiment in the BS211 laboratory of the Department of Environmental science and Engineering, National Pingtung University of Science and Technology, Taiwan.

**Conflicts of Interest:** The authors declare no conflict of interest.

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
