# Peer review of "Occurrence and Degradation of Free and Conjugated Estrogens in a River Receiving Feedlot Animal Discharge"

_applsci, doi:10.3390/app122311961_

Round 1
Reviewer 1 Report
Publications dealing with river, groundwater or drinking water pollution are very much needed.
However data presentation in the manuscript: “Occurrence and degradation of free and conjugated estrogens in a river receiving feedlot animal discharge” should be adjusted for publication.
The Results and even Material and methods sections are interrupted by discussion and many references (e.g. line 185, 192, 193, 196, 198, 209, 212, 215, 230 and much more), although the discussion is presented once again in a separate Discussion section. This is confusing because it is not clear whether it is describing own or foreign data. Moreover, data presentation is mainly in the form of tables or numbers in the text, which are not illustrative and attractive.
I recommend
- consistently separating the results and discussion sections
-convert some tables (and numbers in the text) into graphs
-better explaining the detection rate. How does a non-detectable concentration correlate with a high detection rate?
-adding the formulas of the studied xenobiotic
-improving the English and the comprehensibility of the text
My question is, is it known if the farm animals were medicated with steroids?
Minor points: All abbreviations (such as COD, DOC, DO, TS etc.) should be explained with the first appearance.
Line 16 please explain detection ratio
Line 109 Please explain COD
Line 112 Please explain DOC
Figure 3 legend Linear relationship but the relationship is not linear whole time
Author Response
NO

Reviewer 2 Report
The aim of this study was to evaluate the occurrence and degradation of free and conjugated estrogens in a river receiving feedlot animal discharge. It is worth emphasizing that this aim is not clear in the abstract nor the methodology used for comparing in situ and laboratory results.
The authors used average, median, boxplot and standard deviation, although without carrying out statistical tests to evaluate it the data have or not Gaussian distribution.
The conclusions are not supported by the results because there are a lot of statements without a consistent statistical approach, as for example:
- "The results indicated the estrogen degradation in situ significantly differed from the laboratory." I did not notice a statistical comparison to support this statement.
- "Although the E1 was more resistant to degradation than E2 and E3 at low concentrations, they rapidly degraded at high ambient concentrations." There is partial overlap between 9.71 +/- 6.42 (E3) and 3.55 +/- 2.41 (E2).
- "Observed from the beginning and 24 h estrogen concentration (Table 3), only free estrogens E1, E2, and E3 concentrations significantly decreased, DES concentration slightly decreased, and E1-3G concentration did not decrease." How can these statements be considered as true?
- "Figure 2 shows DES and E1-3G average concentrations (Cn/C0) changed with time at six sites." Once more, there are partial overlap of error bands and, unfortunately, I cannot agree with your conclusions.
- "Figure 3 shows three free estrogen average concentrations (ln Cn/C0) changed with time at six sites and shows linear regression...". Please, explain clearly, based on statistical tests why a linear regression was used rather a quadratic one, for example.
Author Response
NO

Round 2
Reviewer 1 Report
The authors have corrected and accepted all suggested changes.
Author Response
Thank you for the comment.
Reviewer 2 Report
The average E1 54.15 ± 31.42 ng/L concentrations were significantly higher than E3 (9.71 ± 6.42 ng/L) and E2 (3.55 ± 2.41 ng/L) concentrations (p< 0.001). Sentence without sense! I suggest withdrawing it.
The normality of the data set has been tested by the Kolmogorov-Smirnov (K-S) method. Why K-S? Up to 50 results, Shapiro Wilk should be used instead of K-S.
Most concentration sets of estrogens are not normal (Gaussian) distribution; hence, the data set presents the average, median and standard deviation, which is shown in Figure 2. The mean and median values are very close. Were outliers treated? If data set are not normal, there is no metrological sense to calculate standard deviation.
In addition, Figure 3 has been modified and it shows that the estrogen concentration ratio (Cn/C0) changed with time and linear regression for the five detected compounds. There is no discussion related to the modification in Figure 3.
The linear regression had been statistically tested and the result is listed in Table R1. Figure 4 has been modified and shows estrogen concentration ratio ln (Cn/C0) changed with time and linear regression for E1, E2, and E3. The linear regression had been statistically tested and the result is listed in Table R2. This part of the manuscript has been improved, but not enough. Please enter the two required F tests to confirm this claim.
Author Response
Comments and Suggestions for Authors
Comment #2-1
The average E1 54.15 ± 31.42 ng/L concentrations were significantly higher than E3 (9.71 ± 6.42 ng/L) and E2 (3.55 ± 2.41 ng/L) concentrations (p< 0.001). Sentence without sense! I suggest withdrawing it.
Response #2-1
Thank you for the suggestion.
In Abstract the following has been modified. “The average concentrations were 54.15 ± 31.42, 9.71 ± 6.42 and 3.55 ± 2.41 ng/L for E1, E2 and E3, respectively.”
In Section 3.2.1. Concentration of free estrogen has been modified as follows. “Average concentration of free estrogen were 54.2 ± 31.4, 9.7 ± 6.4 and 3.6 ± 2.4 ng/L for E1, E3 and E2, respectively.”
In Section 4.1. Estrogen concentrations has been modified as follows. “Average concentration of free estrogen were 54.2 ± 31.4, 9.7 ± 6.4 and 3.6 ± 2.4 ng/L for E1, E3 and E2, respectively, which is similar to a polluted river and higher than most rivers' concentrations [48-52].”
Comment #2-2
The normality of the data set has been tested by the Kolmogorov-Smirnov (K-S) method. Why K-S? Up to 50 results, Shapiro Wilk should be used instead of K-S.
Response #2-2
Thank you for the suggestion.
In the present study, the concentrations < MDL were ignored in the statistic calculation; hence, the numbers of estrogen concentration data sets ranged from 11 to 75. Mishra et al. (2019) pointed out the Shapiro–Wilk (S-W) test is the more appropriate method for small sample sizes (<50 samples), although it can also be used on larger sample sizes, while the Kolmogorov–Smirnov (K-S) test is used for n ≥50. Therefore, we used both S-W and K-S test methods to test the normality of the estrogen concentrations data sets. Table 2 has been modified as follows. The p values are deleted for the concentration < MDL; they were considered as 0. The p values were added by the Shapiro Wilk test method.
3.2. Estrogen concentrations in water samples has been modified as follows.
Fig. 2 shows the average concentration and median for each species. The normality test had same results for both test methods. The estrogen concentrations are not in the normal distribution (p< 0.05) for E2, DES, EE2, E1-3G and E2-3G. The E1, E3, E1-3S and E2-3S concentration had a normal distribution.
In addition the statistics methods were added to Section 2.4.3
2.4.3. Statistics
SPSS software was used for statistical analysis. The Gaussian distribution of con-tinuous variables was tested by the Shapiro-Wilk (S-W) and Kolmogorov–Smirnov (K-S) test methods. Furthermore, the linear regression was performed by simple regression analysis.
Table 2. Estrogen concentrations ranges, detection rates and p values of the normality test
Compounds |
Concentration Range (ng/L) |
Detection rate (%) |
p value * |
p value ** |
E1 |
2.0 - 145.6 |
100 |
0.169 |
0.200 |
E2 |
ND - 13.2 |
84 |
0.003 |
<0.001 |
E3 |
ND - 34.0 |
95 |
0.083 |
0.200 |
DES |
ND - 10.0 |
63 |
<0.001 |
<0.001 |
EE2 |
ND - 4.4 |
15 |
0.001 |
<0.001 |
E1-3S |
ND - 13.2 |
76 |
0.455 |
0.131 |
E2-3S |
ND - 10.4 |
71 |
0.053 |
0.098 |
E1-3G |
ND - 10.0 |
56 |
0.001 |
0.005 |
E2-3G |
ND - 3.6 |
15 |
0.005 |
<0.001 |
Normality test, the concentration < MDL was ignored, * first p value Shapiro Wilk (S-W) test method; ** Second p value using Kolmogorov-Smirnov (K-S) test method.
Comment #2-3
Most concentration sets of estrogens are not normal (Gaussian) distribution; hence, the data set presents the average, median and standard deviation, which is shown in Figure 2. The mean and median values are very close. Were outliers treated? If data set are not normal, there is no metrological sense to calculate standard deviation.
Response #2-3
Thank you for the comment and suggestion.
In statistics, removing outlier values is one of the methods to deal with outliers. In the environment, estrogen concentration is highly variable (Chen et al., 2010; Hung et al., 2022). It may be a particular case if it finds unreasonable outliers, which does not mean it is an unreasonable value. Therefore, the concentration < MDL was ignored in the statistic calculation. In addition, we have calculated the relative errors of mean and median values [(mean-median)/mean * 100%]. The relative errors ranged from 6.4% to 20.0%. The low relative error suggests no outliers in the present concentration data sets.
In the modified Fig. 2 the standard deviations are removed.
Comment #2-4
In addition, Figure 3 has been modified and it shows that the estrogen concentration ratio (Cn/C0) changed with time and linear regression for the five detected compounds. There is no discussion related to the modification in Figure 3.
Response #2-5
Thank you for the comment and suggestion.
The discussion of the estrogen concentration ratio (Cn/C0) changed with time. The linear regression for the five detected compounds has been added following Section.
3.3. Degradation of estrogen in the water sample
Fig. 3 shows the average estrogen concentration ratios (Cn/C0) following time and line regression curves. Table 4 lists the linear regression parameters including slope, r value, and p value from 0 – 12 h. The p values (p< 0.05) show E1, E2, and E3 had significantly decreased, while DES and E1-3G had insignificantly decreased. Moreover, DES had a median r value (r=0.59) and a negative slope suggesting DES slightly decreased following time. However E1-3G had a small positive slope and r value > 0. 5, suggesting E1-3G did not change following time.
4.2. Degradation of DES and E1-3G
This study investigated the degradation of estrogens in in-situ river water. Theoretically, the concentration of estrogen will decrease following time if the degradation occurs; hence, the estrogen concentration ratio (Cn/C0) changed with time, having a negative slope and a p value of <0.05 in the linear regression. Fig. 3 shows the degradation curves of DES, E1-3G, E1, E2 and E3. The linear regression had been statistically tested and the result is listed in Table 4. The linear results show E1-3G had a positive slope value (0.0078) and insignificantly decreased (r=0.31, p=0.56). DES had a negative slope value (-0.016) and insignificantly decreased (r=0.60, p=0.21). The E1, E2, and E3 had negative slope values and significantly decreased (p< 0.05) as shown in Table 4. The negative slope value and p< 0.05 suggests the concentration of E1, E2 and E3 significantly decreased following time.
Comment #2-5
The linear regression had been statistically tested and the result is listed in Table R1. Figure 4 has been modified and shows estrogen concentration ratio ln (Cn/C0) changed with time and linear regression for E1, E2, and E3. The linear regression had been statistically tested and the result is listed in Table R2. This part of the manuscript has been improved, but not enough. Please enter the two required F tests to confirm this claim.
Response #2-5
Thank you for the comment and suggestion.
Table 4 (original Table R1) and Table 5 (original Table R2) have added the F-values.
The discussion of the estrogen concentration ratio ln (Cn/C0) changed with time and linear regression for E1, E2, and E3 has been added to Section 4.3.
4.3. Degradation of free estrogen
From the results in Fig. 3 and Table 4, E1, E2 and E3 had significantly decreased following time. According Eq. 1, Fig. 4 shows the average estrogen concentration ratio ln (Cn/C0) changed with time and linear regression for E1, E2, and E3. The linear regression has been statistically tested and the result is listed in (Table 5). For each site, the statistic results of ln (Cn/C0) changing with time, are listed in Table 6. The linear regression of the average ln (Cn/C0) following time had a statistical significantly linear relationship (p= 0.006 - 0.001) for E1, E2 and E3 (Fig. 4).
Table 4. The parameters of linear regression of average Cn/C0 ratios changed with time for the five detected compounds.
parameters |
DES |
E1-3G |
E1 |
E2 |
E3 |
Slope |
-0.016 |
0.0078 |
-0.0714 |
-0.0276 |
-0.058 |
Intercept |
1.030 |
1.011 |
1.009 |
1.007 |
0.9093 |
r |
0.60 |
0.31 |
0.996 |
0.927 |
0.963 |
p-value |
0.212 |
0.56 |
<0.001 |
0.008 |
0.002 |
F-value |
2.20 |
0.41 |
473.06 |
24.47 |
8.34 |
p< 0.05 indicated the data set had a significantly linear relationship. F=3.45 at p=0.05.
Table 5. The parameters of linear regression of average ln Cn/C0 changed with time for E1, E2 and E3
Parameters |
E1 |
E2 |
E3 |
Slope |
-0.154 |
-0.034 |
-0.102 |
Intercept |
0.203 |
0.0197 |
-0.025 |
r |
0.964 |
0.937 |
0.99 |
p-value |
0.002 |
0.006 |
<0.001 |
F-value |
52.15 |
29.10 |
9.52 |
P < 0.05 indicated the data set had significantly linear relationship. F=3.45 at p=0.05.

Round 3
Reviewer 2 Report
Almost all of my comments and suggestions have been addressed. Thank you!
I only have one more suggestion. Please, if possible, include in Tables 4 and 5, the second F-test MSLOF / MSPE (Mean Square of lack of fit / Mean Square of pure error). I suppose that the F-test that was included is MSREG / MSR (Mean square of the regression / Mean square of the residuals).
Author Response
Response #2-1
Thank you for the suggestion.
Table 4 and Table 5 have been added the second F test value (F2, Mean Square of lack of fit/Mean Square of pure error).
Table 4. The parameters of linear regression of average Cn/C0 ratios changed with time for the five detected compounds.
parameters |
DES |
E1-3G |
E1 |
E2 |
E3 |
Slope |
-0.016 |
0.0078 |
-0.0714 |
-0.0276 |
-0.058 |
Intercept |
1.030 |
1.011 |
1.009 |
1.007 |
0.9093 |
r |
0.60 |
0.31 |
0.996 |
0.927 |
0.963 |
p-value |
0.212 |
0.56 |
<0.001 |
0.008 |
0.002 |
F1 |
2.20 |
0.41 |
473.06 |
24.47 |
8.34 |
F2 |
1.10 |
0.67 |
0.05 |
0.26 |
1.64 |
p< 0.05 indicated the data set had a significantly linear relationship.
F1: F-test value is MSREG/MSR (Mean square of the regression/Mean square of the residuals).
F2: F-test value is MSLOF/MSPE (Mean Square of lack of fit/Mean Square of pure error).
Table 5. The parameters of linear regression of average ln Cn/C0 changed with time for E1, E2 and E3
Parameters |
E1 |
E2 |
E3 |
Slope |
-0.154 |
-0.034 |
-0.102 |
Intercept |
0.203 |
0.0197 |
-0.025 |
r |
0.964 |
0.937 |
0.99 |
p-value (R) |
0.002 |
0.006 |
<0.001 |
F1 |
52.15 |
29.10 |
9.52 |
F2 |
0.20 |
0.16 |
0.67 |
p< 0.05 indicated the data set had a significantly linear relationship.
F1: F-test value is MSREG/MSR (Mean square of the regression/Mean square of the residuals).
F2: F-test value is MSLOF/MSPE (Mean Square of lack of fit/Mean Square of pure error).
